# Do Patient-Important Outcomes Differ by Care Setting? Findings from Semi-Structured Interviews with Individuals with Diabetes

**DOI:** 10.3390/healthcare13233116

**Published:** 2025-12-01

**Authors:** Amy T. Cunningham, Alexzandra T. Gentsch, Pouya Arefi, Judd E. Hollander, Marianna D. LaNoue, Amanda M. B. Doty, Geoffrey D. Mills, Brendan Carr, Kristin L. Rising

**Affiliations:** 1Department of Family and Community Medicine, Sidney Kimmel Medical College, Thomas Jefferson University, Philadelphia, PA 19107, USA; geoffrey.mills@jefferson.edu; 2Department of Emergency Medicine, Sidney Kimmel Medical College, Thomas Jefferson University, Philadelphia, PA 19107, USA; alexzandra.gentsch@jefferson.edu (A.T.G.); judd.hollander@jefferson.edu (J.E.H.); amanda.doty@jefferson.edu (A.M.B.D.); kristin.rising@jefferson.edu (K.L.R.); 3Department of Internal Medicine, Icahn School of Medicine at Mount Sinai, New York, NY 10029, USA; pouyaarefi@gmail.com; 4School of Nursing, Vanderbilt University Medical Center, Nashville, TN 37240, USA; marianna.lanoue@vanderbilt.edu; 5Department of Emergency Medicine, Icahn School of Medicine at Mount Sinai and the Mount Sinai Health System, New York, NY 10029, USA; brendan.carr@mountsinai.org

**Keywords:** diabetes, patient-reported outcomes, patient priorities qualitative research

## Abstract

**Background:** Patient-important outcomes (PIOs) reflect patient values and preferences. Prior studies have elicited a variety of PIOs for diabetes. However, no studies have examined whether, or how, PIOs differ across diabetes care settings. The purpose of this study was to compare the frequencies of PIOs derived from patients with diabetes in primary care (PC), acute care (emergency department (ED)), and post-acute care (post-hospital discharge (PHD)) settings within a large delivery system. **Methods:** This study was an analysis of 89 interviews with patients in PC, ED, and PHD settings. Participants had moderately to poorly controlled diabetes, defined as follows: presented to the ED with a diabetes-related problem, admitted to the hospital for a diabetes-related problem, or had at least two primary care measurements of hemoglobin A1c (HbA1c) > 7.5 in the prior year. A matrix analysis compared the frequencies of participants’ PIOs across the three settings. **Results:** Overall PIO frequencies were similar across care settings. PIOs fell into seven domains; all seven domains and 21 of the 26 PIOs were represented within each of the care settings. The most common PIOs included “be healthy”, “eat right”, and “reduce or get off medicines”. **Conclusions:** Participants identified similar PIOs in all care settings, indicating that recruitment from one or two care settings may often be sufficient for achieving saturation of PIOs. Furthermore, the results inform our understanding of patient priorities across the care continuum.

## 1. Introduction

As the healthcare system moves towards more patient-centered care, patient engagement is important to identify patient-important outcomes (PIOs). Patient-reported outcomes (PROs) have been more commonly used as complements to clinical outcomes. In contrast, PIOs are outcomes grounded in individuals’ values and preferences [1,2,3]. For example, traditional diabetes PROs have included treatment satisfaction and quality of life [4,5], while diabetes PIOs have included novel measures such as stopping medications and finding a cure for diabetes [6,7,8].

An underexplored area of research is whether PIOs differ for patients recruited from different care settings, e.g., different environments where healthcare is provided. Historically, the U.S. healthcare system has been fragmented across many dimensions. In order to make this system more patient-centered, there have been calls to envision the health system as a continuum of care focused on patients’ experiences as they journey across different care settings rather than within just one setting [9,10].

The healthcare continuum occurs in several settings: wellness (prevention), pre-acute (monitoring, typically in primary care), acute (emergency or inpatient care), post-acute (post-emergency department/hospital discharge), and homecare (self-management) [11]. Patient experiences and priorities across the care continuum may vary based on their diagnosis, current health status, and illness trajectory. In particular, individuals with chronic illnesses such as diabetes will cycle between different illness periods. These periods include times of relative stability in their health status when they will engage mainly in primary care and self-management. At times, patients will transition from stability to periods of acute exacerbation and symptoms, during which they may utilize the emergency department and/or inpatient care. They will then transition back to periods of stability, forming a “circle of care” [12]. These changes may lead to shifting perspectives in which an individual’s illness moves between the foreground and background of their daily life [13]. These shifts in illness perspective may also lead to different patient priorities depending on their current health status and subsequent care setting.

Qualitative methods, particularly focus groups and interviews, have been extensively used to examine PIOs for chronic conditions [14], for example, examining patients’ general and self-management priorities within one care setting, such as primary care [7,8,15] or a community-based organization [16,17,18,19].

One study examined goal-setting and priorities among patients with diabetes transitioning from inpatient to outpatient care. The authors found that a minority of inpatient nurses established outpatient goals with patients, and these goals were inconsistently communicated to outpatient care teams [17]. To our knowledge, no studies have examined whether, or how, diabetes or other chronic illness PIOs differ across patients sampled from different healthcare settings. This information is vital to inform the design of future studies, as engagement of patients in certain healthcare settings may be easier than others, and data are needed to inform how PIOs change based upon the engagement setting.

In this study, we report a comparison of PIOs derived from patients with diabetes in primary care (PC), acute care (emergency department (ED)), and post-acute care (post-hospital discharge (PHD)) settings within a large delivery system. The findings offer insights to qualitative researchers on whether care setting and associated health status changes may influence PIOs. Furthermore, the results will inform our understanding of patient priorities across the care continuum, supporting more patient-centered goal-setting and care.

## 2. Materials and Methods

This is a secondary analysis of individual qualitative interview data from a study that compared the comprehensiveness and efficiency of semi-structured interviews to group concept mapping for eliciting patient-important diabetes outcomes. The full methods and results of this IRB-approved parent study have been described elsewhere [20]. The aim of this analysis is to identify how PIOs collected from three different health settings during the parent study (PC, ED, PHD) varied.

### 2.1. Sample

Individuals from an urban academic medical center in Philadelphia, PA, were eligible if they were age 18 or older, had a diagnosis of type 1 or type 2 diabetes, spoke English, and could provide informed consent. All participants had moderately to poorly controlled diabetes defined as one of the following: presented to the ED with a diabetes-related problem, admitted to the hospital for a diabetes-related problem, or had at least two primary care measurements of hemoglobin A1c (HbA1c) > 7.5 in the prior year. The definition of moderately to poorly controlled diabetes was developed by several primary care and emergency medicine physicians who were members of the study team.

The entire study was conducted in close collaboration with a Patient and Key Stakeholder Advisory Board (PAKSAB). The PAKSAB included patients, advocates, and providers from diverse care pathway domains. Two members of the PAKSAB were research team members and participated in interview development, data collection, and analysis.

### 2.2. Data Collection

Individuals were identified using electronic medical records (EMR). All participants provided informed consent and were compensated $25. Research approval was obtained from Thomas Jefferson University’s Institutional Review Board. Demographic and medical information was collected by self-report from participants and an EMR review.

An open-ended, semi-structured interview guide was used to discuss outcomes most important to participants when making diabetes management decisions. The study team collaborated with the PAKSAB to develop the interview guide, which was further refined after a mock interview at a PAKSAB meeting. Interview questions asked were about participants’ beliefs (“What do you think diabetes is doing to your body?”), their diabetes care challenges (“When thinking about your diabetes, what worries or concerns do you have?”), and their care goals (“What are your goals for your treatment?”) [21]. Interviews were conducted in person or over the phone, audio recorded, and transcribed professionally with identifying information removed. Interviews were conducted in each healthcare setting until thematic saturation was reached.

### 2.3. Analyses

Transcripts were analyzed using NVivo 11.0 (Denver, CO, USA). Interview analysis for the parent study was conducted using a conventional content analysis approach [22]. Three members of the research team independently coded interviews and one third were randomly double coded throughout the coding process to assess and confirm interrater reliability. We used standards for double-coding by O’Connor & Joffe [23] and standards by Cohen [24] and McHugh [25] to set thresholds for kappa and percent agreement that would determine sufficient inter-coder reliability. Our minimum κ coefficient was 0.6, and our target was 0.8. Our average percentage of agreement was 97%; the research team developed a codebook by independently coding a subsection of transcripts within each setting. Coders analyzed the remaining transcripts with the established codebook, meeting frequently to review coding, resolve discrepancies, and refine the codebook iteratively.

To identify discrete goals, we further analyzed data coded to the “goals” code. The study team worked with the PAKSAB to categorize the identified goals from this code into overarching PIOs and broader domains. This resulted in 26 PIOs across 7 domains [26]. We then performed a descriptive analysis of the PIOs using a matrix analysis to compare PIO frequencies. Comparison of coding frequencies is a well-established approach for analyzing large qualitative datasets [27]. A matrix was created with domains, PIOs, and goals in rows and the healthcare setting in which the participants were engaged during research recruitment (e.g., PC, ED, and PHD) in columns. The number of participants who mentioned a PIO in interviews were calculated for each care setting.

## 3. Results

Demographics of the 89 interview participants can be found in Table 1 [20]. Mean age was 54.6 years. Most participants were Black (68.0%), female (55%), and completed high school as their highest level of education (76.0%). Nearly all (95.5%) had type 2 diabetes, and 83.0% had lived with diabetes for more than five years. The mean HbA1c was 10.2 percent.

Table 2 reports the frequency of PIOs by care setting, organized by PIO domain. Overall, PIO frequencies were similar across care settings. All seven domains and 21 of the 26 PIOs were represented within each of the care settings. The following provides details of results, by domain.

Within the “Achieve measurable goals” domain, participants across all care settings cited three of the PIOs—“Control daily blood sugar”, “Control diabetes”, and “Lose weight”—as important care goals, with controlling blood sugar and diabetes, overall, being mentioned slightly more frequently by ED and PHD participants. One PIO—“Achieve A1c goals”—arose in the PC and ED settings, though not PHD.

PIOs within the domains of “Best utilize medical/professional services” and “Learning about diabetes” were less frequently identified overall. The PIO mentioned most often was “Minimize hospital, emergency dept., and doctor visits”, particularly among ED and PHD participants.

The domains of “Manage diet” and “Manage medications” both occurred frequently across health settings. Participants in all settings commonly cited “Eat right” as a PIO in the “Manage diet” domain. Within “Manage medications”, the most common PIO across all three settings was “Reduce or get off medicines”, followed by “Eliminate injections”.

“Optimize daily self-care” is the domain with the most PIOs, including the broad, widely mentioned PIO “Be healthy,” and other common PIOs including “Be active”, and “Live like a person who does not have diabetes”. There were a few PIOs with minimal use that did not occur in every setting, including “Access alternative treatments” (not in PC or PHD), “Establish a routine” (not in PC), “Minimize symptoms” (not in PC or PHD), and “Reduce stress” (not in PHD).

Finally, in the “Optimize long-term health” domain, all PIOs were represented across all settings. “Get rid of diabetes”, “Prevent complications”, and “Live longer” were frequently mentioned in all settings, although “Get rid of diabetes” and “Prevent complications” were more frequently voiced in the ED and PHD groups, and “Live longer” was more common in the ED group.

The ten most common PIOs across settings are reported in Table 3. The most common PIOs overall were “Be healthy”, “Control diabetes”, and “Control daily blood sugar”.

## 4. Discussion

To our knowledge, this is one of the first studies to compare PIOs by recruitment setting. In our matrix analysis of interviews with patients with primarily type 2 diabetes recruited from PC, ED, and PHD settings, we found that participants across settings reported similar PIOs within seven common domains. All domains and the majority of the PIOs were represented within each health setting.

When looking at the PIOs that were not included across each health setting, there are a few critical findings. While “Achieve A1c goals” (within the “Achieve measurable goals” domain) was not included in PHD, “Control diabetes” (general/overall control) was one of the most frequent PIO among PHD patients. As HbA1c is a primary marker of disease control, this concept was arguably still captured within the PHD setting. The other six PIOs that were not included in all settings had very few references, suggesting that they were of less universal importance. There were three PIOs—“Maximize health insurance”, “Get on an insulin pump”, and “Establish a routine”—that were missing in one setting and only had one reference each in the other two settings. Similarly, there was one PIO—“Access alternative treatments”—that was missing in two settings (PC and PHD) and had only two references in ED. The final two PIOs had the most variability with overall low frequencies. These were “Minimize symptoms”, which had five references in the ED though they did not arise in PC and PHD, and “Reduce stress” which did not arise in PHD though had one reference in PC and three in the ED.

Overall, the most common PIOs within all settings fell under the domains of “Achieve measurable goals”, “Manage diet”, “Manage medications”, “Optimize daily self-care”, and “Optimize long-term health”. Some differences did emerge in the frequencies of PIO mentioned for PC. For example, ED and PHD patients more frequently mentioned wanting better blood sugar control and general diabetes control, reducing medical visits, preventing complications, and getting rid of diabetes. These findings may reflect that these patients had recent emergency or inpatient care for their diabetes, so they may feel less control over their illness and have more worries about complications than those patients with diabetes recruited from primary care. However, these outcomes were all mentioned by PC participants, so they are still relevant across care locations.

Our findings further add to the literature on qualitative participant recruitment by comparing PIOs from patients recruited across a range of care settings. We demonstrate that the vast majority of PIOs, and the most common PIOs, arose from participants in all three care settings. These findings suggest that for studies that involve eliciting perspectives from participants across care settings or the continuum of disease, recruitment from one or two care settings may often be sufficient for achieving saturation of PIOs. This could allow researchers to increase study efficiency while still collecting rich data.

Our study has limitations. First, our analysis was a secondary analysis of qualitative interview data. Although the analysis reported in this manuscript was not part of the original study’s objectives, it utilized coding generated during the primary study in which multiple steps were taken to ensure rigor in the coding process, including multiple coders, an audit trail, and member checking through the review of findings by our Patient and Key Stakeholder Advisory Board (PCORI final research report). Additionally, interviews were conducted either by phone or in person, which may have affected participant responses. Our study also included individuals with both type 1 and type 2 diabetes, which can have different disease trajectories and potentially different PIOs. Finally, our participants consisted of a convenience sample of patients in one health system, most of whom were Black and female, and consisted primarily of patients with type 2 diabetes. Our sample limits the generalizability of our findings; a broader range of health systems and demographic categories is ideal, and PIOs for other health conditions should also be examined. Nevertheless, the findings provide valuable information on similarities and differences in diabetes PIOs across primary care, emergency care, and post-hospital discharge.

## 5. Conclusions

Participants identified similar PIOs for diabetes in all care settings, indicating that recruitment from one or two care settings may often be sufficient for achieving saturation of PIOs. Furthermore, the results inform our understanding of patient priorities across the care continuum. By eliciting patient priorities, this work support the design of further patient-centered research and care.

## Figures and Tables

**Table 1 healthcare-13-03116-t001:** Participant Demographics (*n* = 89).

Characteristic	*n* (%) *
Age, mean (SD)	54.6 (13.8)
Ethnicity	
Hispanic/Latino	8 (9)
Not Hispanic/Latino	80 (90)
Race	
White	24 (27)
Black	60 (68)
Asian	2 (2)
Other	3 (3)
Sex	
Male	40 (45)
Female	49 (55)
Education	
Less than High School	4 (5)
High school graduate	68 (76)
College Degree	13 (15)
Post-Grad degree	4 (5)
Annual Household Income	
<10 K	15 (21)
10–25 K	22 (31)
25–50 K	19 (27)
50–99 K	7 (10)
>100 K	8 (11)
Insurance Type (could have >1)	
Medicaid	38 (43)
Medicare	37 (42)
VA/DOD	2 (2)
Private Insurance	31 (35)
Uninsured	9 (10)
Type 1 Diabetes	4 (4.5)
Type 2 Diabetes	85 (95.5)
Mean HbA1c %	10.2 (3.3)
Type 2 Diabetes Insulin Use	60 (70)
Body Mass Index, mean (SD)	34.8 (10.3)
Hospital Admits in past 12 months, mean (SD)	2.3 (4.1)
ED Visits in past 12 months, mean (SD)	2.8 (4.3)
Doctor Visits in past 12 months, mean (SD)	11.2 (4.3)
Years since diagnosis	
<1 year	2 (2)
1–5 years	12 (13)
>5 years	74 (83)
Health status (mean, SD) (range 1–5: 1 = excellent and 5 = poor)	3.6 (0.9)

* Note: Participants were not required to respond to all questions; therefore, some percentages may not be equal to 100.

**Table 2 healthcare-13-03116-t002:** Frequency of patient-important goals, outcomes, and domains by care setting.

Domain	Patient-Important Outcome (PIO)	Goals	Primary Care (PC) (*n* = 30)	Post-Acute Care (PHD) (*n* = 29)	Acute Care (ED) (*n* = 30)
**Achieve Measurable Goals**	**Achieve A1c goals**		**5**	**0**	**7**
	**Control daily blood sugar**		**8**	**16**	**14**
		Control or lower blood sugar	8	15	14
		Prevent DKA	0	2	0
		Prevent hyperglycemia	0	1	1
		Prevent hypoglycemia	0	0	1
	**Control diabetes**		**9**	**17**	**13**
		Obtain a good report from doctor	1	2	0
		Maintain current management	3	4	1
		Manage diabetes better	4	8	11
		Prevent bad news	0	0	1
		Prevent diabetes or health from becoming worse	1	3	0
	**Lose weight**		**8**	**11**	**7**
**Best Utilize Medical/Professional Services**	**Maximize health insurance**		**1**	**1**	**0**
	**Minimize hospital, emergency dept., and doctor visits**		**1**	**3**	**5**
**Learning about Diabetes**	**Learn about diabetes**		**2**	**1**	**1**
	**Participate in peer support**		**2**	**1**	**1**
		Be an inspiration to others	0	1	1
		Continue or start diabetes group	2	0	0
**Manage Diet**	**Eat right**		**10**	**10**	**12**
	**Understand how to eat well with diabetes**		**2**	**1**	**1**
**Manage Medications**	**Eliminate injections**		**3**	**4**	**7**
		Avoid insulin or injections	0	1	3
		Get off insulin or injections	3	3	4
	**Get on an insulin pump**		**1**	**0**	**1**
	**Reduce or get off medicines**		**13**	**7**	**10**
		Control blood sugars naturally	0	2	0
		Get off medicine	9	5	7
		Prevent more meds	3	0	0
		Reduce meds	7	4	3
**Optimize daily self-care**	**Access alternative treatments**		**0**	**0**	**2**
	**Be healthy**		**17**	**21**	**16**
		Alleviate illness	2	0	0
		Avoid unhealthy things	0	0	1
		Be healthy	16	20	15
		Be stable	0	2	0
		Control blood pressure	1	1	1
		Control cholesterol	1	0	1
		Get rid of A fib	1	0	0
		Prevent illness	0	5	1
		Prevent pancreatitis flare up	0	0	1
		Quit smoking	0	0	1
		Stop drinking alcohol	0	0	1
		Take better care of myself	0	0	1
		To feel good	0	3	0
		To feel rested	0	1	0
		To sleep well	0	1	0
	**Be more active**		**10**	**10**	**9**
		Be active with family	2	0	1
		Develop more hobbies	0	4	1
		Get in shape	0	3	0
		Improve mobility, stamina, ability to do things	2	6	4
		Improve strength	0	1	1
		Increase physical activity or stay active	7	5	6
	**Establish a routine**		**0**	**1**	**1**
		Get organized	0	1	0
		Have a set schedule	0	1	1
	**Have a personalized care plan**		**5**	**2**	**7**
		Consistently monitor blood sugar	0	0	1
		Follow doctor’s treatment	0	0	2
		Monitor condition	0	1	0
		See doctors regularly or start doctor visits	2	1	2
		Take meds as directed	4	1	2
		Test blood as directed	0	0	1
	**Improve mental health**		**4**	**5**	**5**
		Be happy	2	3	0
		Become better person	0	0	2
		Control mindset or self control	1	3	3
		Prevent depression	1	0	1
	**Live like a person who does not have diabetes**		**7**	**8**	**5**
		Have a normal life	1	1	0
		Have children	1	0	0
		Improve overall lifestyle	4	1	2
		Take care of family	3	6	4
	**Minimize symptoms**		**0**	**0**	**5**
		Get rid of rash	0	0	1
		Manage pain	0	0	2
		Prevent a rash	0	0	1
		Prevent pain	0	0	2
	**Reduce stress**		**1**	**0**	**3**
**Optimize long-term health**	**Get rid of diabetes**		**4**	**10**	**13**
	**Live longer**		**7**	**6**	**13**
		Live longer with family	1	1	2
		Prevent death	6	5	12
	**Maintain independence in personal care**		**2**	**2**	**2**
		Be independent	2	1	2
		Maintain ability to use restroom	0	0	1
		Prevent mobility issues	0	1	0
	**Prevent complications**		**9**	**15**	**12**
		Get rid of neuropathy	1	0	1
		Improve vision or retinopathy	1	1	0
		Manage neuropathy	0	0	1
		Prevent amputation	4	8	8
		Prevent blindness (and issues with eyesight) or retinopathy	5	3	4
		Prevent deterioration or damage to body	0	2	0
		Prevent dialysis	3	3	1
		Prevent heart attack and heart disease	3	1	4
		Prevent infections and wounds	0	0	1
		Prevent kidney problems	1	4	1
		Prevent stroke	0	0	1

**Table 3 healthcare-13-03116-t003:** Ten most common PIOs across settings.

	Number and Percentage of PIO Mentions Within Settings			Number and Percentage of PIO Mentions Across Settings
Patient-Important Outcome (PIO)	Primary Care (PC) (N = 30)	Post-Acute Care (PHD) (N = 29)	Acute Care (ED) (N = 30)	Total Participants (N-89)
**Be healthy**	17 (3.6)	21 (6.9)	16 (5.0)	54 (6.2)
**Control diabetes**	9 (3.2)	17 (5.6)	13 (4.0)	39 (4.5)
**Control daily blood sugar**	8 (3.6)	16 (5.3)	14 (4.4)	38 (4.3)
**Prevent complications**	9 (4.0)	15 (4.9)	12 (3.7)	36 (4.1)
**Eat right**	10 (5.2)	10 (3.3)	12 (3.7)	32 (3.7)
**Reduce or get off medicines**	13 (4.0)	7 (2.3)	10 (3.1)	30 (3.4)
**Be more active**	10 (1.6)	10 (3.3)	9 (2.8)	29 (3.3)
**Get rid of diabetes**	4 (3.2)	10 (3.3)	13 (4.0)	27 (3.1)
**Lose weight**	8 (2.8)	11 (3.6)	7 (2.2)	26 (3.0)
**Live longer**	7 (2.8)	6 (2.0)	13 (4.0)	26 (3.0)

## Data Availability

The datasets used and analyzed during the current study are available from the corresponding author on reasonable request.

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
