# Peer review of "Do Patient-Important Outcomes Differ by Care Setting? Findings from Semi-Structured Interviews with Individuals with Diabetes"

_healthcare, 2025, doi:10.3390/healthcare13233116_

Round 1
Reviewer 1 Report
Comments and Suggestions for Authors
Although this study is defined as a secondary analysis, the limitations of this approach should be adequately discussed and stated in the article. Because the sampling method relied on a convenience sampling approach, most participants were from the same healthcare system and had similar sociodemographic characteristics, limiting generalizability. The definition of "moderately or poorly controlled diabetes" was based solely on the criteria of HbA1c > 7.5 or emergency admission, but the validity of these criteria and their inclusion of different clinical variations were not methodologically justified. Semi-structured interviews were conducted during the data collection process. Three members of the same research team coded, and only 33% were coded twice, indicating limited inter-coder reliability in terms of qualitative data. Please explain.
Author Response
Response to Reviewers
Reviewer 1:
- Although this study is defined as a secondary analysis, the limitations of this approach should be adequately discussed and stated in the article.
Thank you for raising this point. We agree that secondary analysis of qualitative data can have limitations. Although the analysis reported in this manuscript was not part of the study’s original objectives, it used coding generated during the primary study in which multiple steps were taken to ensure rigor in the coding process, including multiple coders, an audit trail, and member checking through review of findings by our Patient and Key Stakeholder Advisory Board. We have acknowledged this limitation and steps we took to ensure rigor in our Discussion section (lines 268-279).
- Because the sampling method relied on a convenience sampling approach, most participants were from the same healthcare system and had similar sociodemographic characteristics, limiting generalizability.
We agree, and have noted this in the limitations section of our discussion (line 282).
- The definition of "moderately or poorly controlled diabetes" was based solely on the criteria of HbA1c > 7.5 or emergency admission, but the validity of these criteria and their inclusion of different clinical variations were not methodologically justified.
We acknowledge that there are varying definitions of “moderately or poorly-controlled diabetes.” Our definitions were developed for the purposes of our study. The criteria for moderately-to-poorly controlled diabetes in each setting were developed by members of the study team who were emergency medicine and primary care physicians. We have clarified this in the Materials and Methods section (lines 116-118).
- Semi-structured interviews were conducted during the data collection process.
Three members of the same research team coded, and only 33% were coded twice, indicating limited inter-coder reliability in terms of qualitative data. Please explain.
We appreciate the need for further detail on our process of double-coding and determining inter-coder reliability. While there is little consensus on what percentage of data should be double coded, O’Connor & Joffe (2020) suggest 10-25% is standard practice. We randomly selected 33% of the data to be double coded throughout the coding process and used standards by Cohen and Mchugh to set thresholds for kappa and percent agreement that would determine sufficient inter-coder reliability. Our minimum κ coefficient was 0.6, and our target was 0.8. Our average percentage of agreement was 97%.; We have added this information to the manuscript (lines 146-150).
Reviewer 2 Report
Comments and Suggestions for Authors
The main question and objective of the study is to analyze and compare the frequencies of PIOs derived from patients with diabetes in primary care (PC), acute care (emergency department (ED)), and post-acute care (post-hospital discharge (PHD)) settings within a large delivery system. The topic is original and relevant to the field. Diabetes is a significant health problem due to the potentially severe consequences it can have on the quality and duration of life if not properly managed. The patient-centered model of health care requires analyzing the experiences and priorities of patients during the different periods of the chronic disease – which is diabetes.
The treatment of diabetes requires not only the administration of insulin or other medications, but also compliance with a certain dietary regimen, sufficient physical activity and self-monitoring of blood sugar levels and management of risk factors that could lead to complications of the condition. Clarifying the PIO in different periods and conditions of the disease would lead to a better understanding, better approach and organization of health care for such patients. There are significant gaps in the understanding of the priorities of chronically ill patients, which can lead to insufficiently relevant health care.
Based on the matrix analysis performed on patients mainly with type 2 diabetes, similar PIOs are reported in seven general areas. Critically, identification goals were also found that were not included in every health environment. This would help in developing an approach and planning of health care for patients with diabetes depending on the phase of their disease.
The methodology of the scientific study, meets the protocol and requirements, the coding methods used show careful consideration of the transcribed data to identify recurring patterns and categories. This process helps in organizing and making sense of the information obtained.
The results are summarized and synthesized according to the set research objective. The study could be used to improve the concept of health care during different periods of chronic illness, but it is necessary to expand the scope and include participants from a wider range of health systems and demographic categories, as noted in the study limitations.
I believe that the references are adequate.
The tables in the scientific study are sufficiently informative.
Regarding Table 1 - row No. 149. It would be good to clarify why the total sum of the percentages for some of the data is not 100. A discrepancy is noted in the distribution by gender.
Since Table 1 was used in a previous study, I believe that it is appropriate to include the relevant reference.
Author Response
Reviewer 2
- The main question and objective of the study is to analyze and compare the frequencies of PIOs derived from patients with diabetes in primary care (PC), acute care (emergency department (ED)), and post-acute care (post-hospital discharge (PHD)) settings within a large delivery system. The topic is original and relevant to the field. Diabetes is a significant health problem due to the potentially severe consequences it can have on the quality and duration of life if not properly managed. The patient-centered model of health care requires analyzing the experiences and priorities of patients during the different periods of the chronic disease – which is diabetes. The treatment of diabetes requires not only the administration of insulin or other medications, but also compliance with a certain dietary regimen, sufficient physical activity and self-monitoring of blood sugar levels and management of risk factors that could lead to complications of the condition. Clarifying the PIO in different periods and conditions of the disease would lead to a better understanding, better approach and organization of health care for such patients. There are significant gaps in the understanding of the priorities of chronically ill patients, which can lead to insufficiently relevant health care. Based on the matrix analysis performed on patients mainly with type 2 diabetes, similar PIOs are reported in seven general areas. Critically, identification goals were also found that were not included in every health environment. This would help in developing an approach and planning of health care for patients with diabetes depending on the phase of their disease.
Thank you for this summary and positive feedback.
- The methodology of the scientific study, meets the protocol and requirements, the coding methods used show careful consideration of the transcribed data to identify recurring patterns and categories. This process helps in organizing and making sense of the information obtained.
Thank you; we appreciate this feedback.
- The results are summarized and synthesized according to the set research objective. The study could be used to improve the concept of health care during different periods of chronic illness, but it is necessary to expand the scope and include participants from a wider range of health systems and demographic categories, as noted in the study limitations.
We agree that future research with more health systems and patients from different demographic categories.
- I believe that the references are adequate.
Thank you.
- The tables in the scientific study are sufficiently informative.
Thank you.
- Regarding Table 1 - row No. 149. It would be good to clarify why the total sum of the percentages for some of the data is not 100. A discrepancy is noted in the distribution by gender.
Thank you for noting this—we have added a note to the table indicating that participants did not respond to all demographic questions, leading the sum of the percentages for some categories to not equal 100 (lines 176-177). Additionally, the Female category should be 49, rather than 9; we have made that correction (line 171).
- Since Table 1 was used in a previous study, I believe that it is appropriate to include the relevant reference.
We have added this reference (line 166).
Reviewer 3 Report
Comments and Suggestions for Authors
Thank you for the opportunity to review this timely manuscript. This manuscript addresses an important and underexplored question in diabetes care by examining whether patient-important outcomes (PIOs) differ across primary, acute, and post-acute care settings. It is well-structured and leverages a robust qualitative dataset. The inclusion of a Patient and Key Stakeholder Advisory Board enhances the study’s patient-centered approach and credibility. Please see the following suggestions to strengthen the manuscript:
- Abstract - Consider adding specific numbers or examples of key PIOs. Stating, "Overall PIO frequencies were similar" is a bit vague. Also consider adding the most common PIOs (e.g., Be healthy).
- Introduction - In the first paragraph of the introduction, please define what you mean by ‘care settings’ earlier in the text. Currently, you refer to different settings (primary care, acute care, post-acute care) before explaining what they are, which can be confusing for readers unfamiliar with the terminology.
- Methods - Please clarify how the interview guide was developed and include examples of the key questions asked. This will help readers understand the basis for your data collection and ensure transparency.
- Results - To help with the interpretation of the results consider creating a smaller table (e.g., top 10 most frequent PIOs across all care settings, with percentages based on sample sizes).
- Discussion -
- Consider adding a greater emphasis on the practical implications for recruitment strategies and patient-centered care. For example, findings suggest that recruiting from one or two care settings may be sufficient to capture the majority of patient-important outcomes (PIOs) which can reduce study complexity and costs without compromising data richness.
- Consider suggesting additional future research ideas such as applying the same methodology to other chronic conditions (e.g., heart failure, COPD) to determine whether PIO patterns differ by disease type.
- Consider acknowledging additional limitations (1) the interviews were conducted both in person and over the phone - these different modes of data collection may have influenced the depth or quality of responses, (2) combining participants with type 1 and type 2 diabetes - these groups often have different disease trajectories, management strategies, and priorities, which could influence patient-important outcomes. Please consider discussing how it might affect interpretation of the findings.
- Conclusion - Consider closing with a stronger action oriented sentence such as.... By highlighting common patient priorities, this work supports the design of patient-centered interventions and policies that reflect what matters most to patients.
Author Response
Reviewer 3:
Thank you for the opportunity to review this timely manuscript. This manuscript addresses an important and underexplored question in diabetes care by examining whether patient-important outcomes (PIOs) differ across primary, acute, and post-acute care settings. It is well-structured and leverages a robust qualitative dataset. The inclusion of a Patient and Key Stakeholder Advisory Board enhances the study’s patient-centered approach and credibility. Please see the following suggestions to strengthen the manuscript:
- Abstract - Consider adding specific numbers or examples of key PIOs. Stating, "Overall PIO frequencies were similar" is a bit vague. Also consider adding the most common PIOs (e.g., Be healthy).
Thank you; we have added specific numbers for the domains, PIOs, and listed several of the most common PIOs (lines 34-37).
- Introduction - In the first paragraph of the introduction, please define what you mean by ‘care settings’ earlier in the text. Currently, you refer to different settings (primary care, acute care, post-acute care) before explaining what they are, which can be confusing for readers unfamiliar with the terminology.
We have defined care settings in the introduction (lines 56-57).
- Methods - Please clarify how the interview guide was developed and include examples of the key questions asked. This will help readers understand the basis for your data collection and ensure transparency.
We have added further information on the interview guide development and have referenced the project’s final research report, which contains the full interview guide (lines 132-137).
- Results - To help with the interpretation of the results consider creating a smaller table (e.g., top 10 most frequent PIOs across all care settings, with percentages based on sample sizes).
We have added this table (lines 180-181).
- Discussion -
- Consider adding a greater emphasis on the practical implications for recruitment strategies and patient-centered care. For example, findings suggest that recruiting from one or two care settings may be sufficient to capture the majority of patient-important outcomes (PIOs) which can reduce study complexity and costs without compromising data richness.
Thank you; we have added a line addressing this to the Discussion (lines 268-269).
- Consider suggesting additional future research ideas such as applying the same methodology to other chronic conditions (e.g., heart failure, COPD) to determine whether PIO patterns differ by disease type.
Thank you; we have added this (line 282-284).
- Consider acknowledging additional limitations (1) the interviews were conducted both in person and over the phone - these different modes of data collection may have influenced the depth or quality of responses, (2) combining participants with type 1 and type 2 diabetes - these groups often have different disease trajectories, management strategies, and priorities, which could influence patient-important outcomes. Please consider discussing how it might affect interpretation of the findings.
We have added these limitations to the Discussion section (lines 276-279).
- Conclusion - Consider closing with a stronger action oriented sentence such as.... By highlighting common patient priorities, this work supports the design of patient-centered interventions and policies that reflect what matters most to patients.
We appreciate this feedback, and have added a sentence to the Conclusion (lines 288-293).
Round 2
Reviewer 1 Report
Comments and Suggestions for Authors
The authors addressed my concerns about the article, it is publishable